# Human cardiac fibroblasts adaptive responses to controlled combined mechanical strain and oxygen changes in vitro

Giovanni Stefano Ugolini[1], Andrea Pavesi[2,3], Marco Rasponi[1], Gianfranco Beniamino Fiore[1], Roger Kamm[3,4], Monica Soncini[1]*

[1]Department of Electronics, Information and Bioengineering, Politecnico di Milano, Milan, Italy; [2]Institute of Molecular and Cell Biology, Agency for Science, Technology and Research, Singapore, Singapore; [3]Biosym IRG, Singapore-MIT Alliance for Research and Technology, Singapore, Singapore; [4]Department of Biological Engineering, Massachusetts Institute of Technology, Cambridge, United States

**Abstract** Upon cardiac pathological conditions such as ischemia, microenvironmental changes instruct a series of cellular responses that trigger cardiac fibroblasts-mediated tissue adaptation and inflammation. A comprehensive model of how early environmental changes may induce cardiac fibroblasts (CF) pathological responses is far from being elucidated, partly due to the lack of approaches involving complex and simultaneous environmental stimulation. Here, we provide a first analysis of human primary CF behavior by means of a multi-stimulus microdevice for combined application of cyclic mechanical strain and controlled oxygen tension. Our findings elucidate differential human CFs responses to different combinations of the above stimuli. Individual stimuli cause proliferative effects (PHH3[+] mitotic cells, YAP translocation, PDGF secretion) or increase collagen presence. Interestingly, only the combination of hypoxia and a simulated loss of contractility (2% strain) is able to additionally induce increased CF release of inflammatory and pro-fibrotic cytokines and matrix metalloproteinases.

*For correspondence: monica.soncini@polimi.it

Competing interests: The authors declare that no competing interests exist.

## Introduction

When supply of oxygen and nutrients to the myocardium is critically reduced (ischemia), a complex tissue response takes place: within hours tissue necrosis and death of contractile cardiac myocytes occurs in the infarcted area giving rise to an inflammatory phase that recruits immune cells and activates quiescent cardiac fibroblasts (CFs); within a few days a proliferative phase begins, where activated CFs invade the infarcted area and contribute to degrading and replacing the extra-cellular matrix with a collagen-based scar; within weeks the maturation of the fibrotic scar is completed (*Frangogiannis, 2014*; *Heusch et al., 2014*). Cellular and molecular events such as excessive proliferation of CFs, phenotypic switch of CFs, high levels of inflammatory cytokines and humoral factors, unbalanced synthesis of extracellular matrix (ECM) proteins and matrix metalloproteinases (MMP)-mediated degradation of ECM are generally regarded as hallmarks of early fibrotic tissue response (*Fan et al., 2012*; *Krenning et al., 2010*; *Porter and Turner, 2009*; *Shinde and Frangogiannis, 2014*). This essential process maintains tissue integrity, however, it often leads to excessive and adverse CFs remodeling of non-infarcted areas (*Fan et al., 2012*; *Kania et al., 2009*; *Talman and Ruskoaho, 2016*) associated with cardiac dysfunction and increased mortality (*Okada et al., 2005*).

**eLife digest** When the supply of oxygen to the heart is reduced, its cells start to die within hours, the heart muscle becomes less able to contract, and the area becomes inflamed. This inflammation is accompanied by an influx of immune cells. It also activates other cells known as cardiac fibroblasts that help to break down the framework of molecules that supported the damaged heart tissue and replace it with a scar. This response is part of the normal repair process, but it can lead to the formation of scar tissue in non-damaged areas of the heart. Excess scar tissue makes the heart muscle less able to contract and increases the affected individual's chance of dying.

Understanding how this repair process works is an important step in developing strategies to minimise the damage caused by coronary artery disease or heart attacks. However, existing laboratory models are only partly able to recreate the conditions seen in real heart tissue. To properly understand the response at the level of living cells, a more complete model is needed.

Ugolini et al. now report improvements to a small device, referred to as a lab-on-chip, that can subject cells to mechanical strain. The improvements mean the device could also recreate other conditions seen early on in damaged heart tissue, specifically the reduced supply of oxygen. Replicating combinations of mechanical changes and oxygen supplies meant that the impact of these conditions on human cardiac fibroblasts could be directly observed in the laboratory for the first time. Ugolini et al. found that a lack of contraction and low oxygen levels triggered the cardiac fibroblasts to produce inflammatory molecules and molecules associated with the formation of scar tissue. This resembles the response seen in living hearts.

The next step is to improve the lab-on-chip device further by adding other cell types, including heart muscle cells and immune cells. A more complete model may aid future research into how our hearts operate in both health and disease.

Controlled anti-fibrotic strategies still require deeper understanding and advanced models of cardiac fibrosis mechanisms (*Leask, 2010*; *Roubille et al., 2014*).

In general, the onset of pathological myocardial conditions causes alterations of specific environmental cues at the cellular scale: mechanical strain decreases (loss of contractility); oxygen and nutrient levels dramatically decrease (ischemia); levels of inflammatory cytokines increase (post-injury inflammatory response). In an attempt to provide in vitro models of cardiac disease, CFs have been widely studied under relevant physico-chemical stimulation such as mechanical stress (*Schroer and Merryman, 2015*; *Tomasek et al., 2002*), oxygen deprivation (*Clancy et al., 2007*; *Tamamori et al., 1997*) and biochemical stimulation with pro-fibrotic cytokines (*Edgley et al., 2012*; *Lijnen et al., 2000*; *Petrov et al., 2002*). The application of mechanical stress has been shown to have the following effects on CFs: increased ECM protein synthesis (*Carver et al., 1991*); controversial proliferative behavior (*Atance et al., 2004*; *Butt and Bishop, 1997*; *Dalla Costa et al., 2010*; *Liao et al., 2004*), with recent reports of strain intensity-dependent effects (*Ugolini et al., 2016*); increased production of pro-fibrotic and inflammatory cytokines such as TGF-$\beta$ (transforming growth factor-beta) (*Leask, 2007*) and TNF-$\alpha$ (tumor necrosis factor-alpha) (*Yokoyama et al., 1999*).

While previous mechanical stress experiments were performed in a normoxic environment (NX, approximately 20% $O_2$, the standard oxygen level of ambient air), CFs were shown to be sensitive to $O_2$ level variations from physoxia (PX, defined as the physiologic oxygen level in living tissues (about 5–6% $O_2$ in the myocardium [*Gonschior et al., 1992*; *Roy et al., 2003*; *Sen et al., 2006*; *Winegrad et al., 1999*]). Both hypoxia (HX, 1–3% $O_2$) and NX induce a pro-inflammatory and fibrogenic phenotype in cultured CFs (*Roy et al., 2003*; *Sen and Roy, 2010*). These findings imply that normoxic oxygen levels are perceived by CFs as a state of hyperoxia and that a significant bias exists when culturing CFs in non-physiological oxygen environments. The exposure of CFs to HX has been shown to induce collagen production and proliferation of CFs (*Gao et al., 2014*; *Tamamori et al., 1997*), while studies suggest that MMP-based remodeling may not be triggered by HX alone (*Riches et al., 2009*).

Given the complexity of in vivo pathological evolution, the elucidation of how environmental stimuli interplay and guide cellular responses is paramount. To date, no comprehensive model is able to

recapitulate early cellular events taking place after acute myocardial injury and there is no report of cardiac cells subjected to simultaneous mechanical strain and controlled oxygen changes, two major environmental variables involved in cardiac injury and CF adaptive responses. This is mostly due to the lack of compact platforms enabling the controlled application of multiple stimuli.

To address this need, we here report the improvement of a previously described multi-chamber microdevice dedicated to the application of cyclic strain to cell monolayers (*Ugolini et al., 2016*). By adding a system for controlling oxygen changes, we significantly expanded the experimental complexity and mimicking capabilities of the device. We here finely controlled the applied mechanical strain and oxygen regimes sensed by CFs to model early environmental changes in cardiac injury and provide insights into individual or synergistic contributions of the environmental signals in the activation of early CF responses relevant to cardiac fibrotic disease. Shortly after an ischemic myocardial insult (e.g., acute coronary artery occlusion) tissue oxygen levels drop from approximately 5–6% $O_2$ (*Gonschior et al., 1992*; *Sen et al., 2006*; *Winegrad et al., 1999*) to near-zero (*Roy et al., 2003*), rapidly inducing loss of cardiac myocytes contractility. In terms of in vitro model parameters, we thus selected 5% $O_2$ (PX) as a physiological oxygen level and 1% $O_2$ (HX) as an oxygen level characteristic of ischemic myocardial injury. Dramatic alterations of injured myocardial tissue also take place in early timeframes: tissue bulging and dilations are commonly observed together with loss of contractile function (*Eek et al., 2010*; *Pfeffer and Braunwald, 1990*; *Picard et al., 1990*; *Tennant and Wiggers, 1935*). Cardiac imaging studies agree in the interpretation that global strains are abruptly reduced shortly after myocardial insult (*Flachskampf et al., 2011*; *Hoit, 2011*; *Mollema et al., 2010*). Quantitatively, strain values recorded vary significantly throughout the heart. Although absolute values may depend on imaging algorithms, strains in ischemic/injured regions have been reported to be 2–4 fold lower than in healthy myocardial regions (e.g. less than 3% in ischemic regions versus approximately 10% in healthy regions [*Dandel et al., 2009*; *Vartdal et al., 2007*]). Within this range and in line with previous in vitro observations of CFs behavior (*Ugolini et al., 2016*), we selected 2% strain as indicative of reduced contractility and 8% strain as representative of full myocardial contractility. These environmental changes and the following initial cellular responses happen in a timeframe of hours, as shown in vitro (*van Nieuwenhoven et al., 2013*; *Turner et al., 2007*, *2009*) as well as in vivo (*Guillén et al., 1995*; *Morishita et al., 2015*). We thus performed experiments lasting 24 hr and evaluated the main early aspects that govern fibrotic responses in the injured myocardium: ECM remodeling, with stainings for collagen I and quantifications of MMP expression; proliferation of CFs, with analyses of mitotic cells, Hippo pathway signaling and mitogenic PDGF (platelet-derived growth factor) expression; secretion of inflammatory and pro-fibrotic cytokines in CF supernatants; myofibroblast differentiation, through stainings for αSMA (α-smooth muscle actin). Results of our investigation provide insights into how the combination of environmental stimuli may act synergistically or independently to drive in vitro adaptive early CF responses.

## Results

### ECM remodeling

Myocardial pathological remodeling is largely based on unbalanced CF production of collagen I and MMP-mediated matrix remodeling. To analyze CF-mediated early remodeling events under combined stimulations, we analyzed immunofluorescence images of CFs stained for collagen I. In addition, we quantified the expression of MMP-2 and MMP-3 in cell culture supernatants, two enzymes specifically expressed by CFs during cardiac remodeling (*Fan et al., 2012*). While CFs normally exhibit diffuse cytoplasmic staining for collagen I, an intense staining localized to perinuclear regions of CFs was observed after 24 hr under specific stimulations (*Figure 1A*). Based on analysis of intracellular fluorescence intensity, we found that the increase in collagen I was similarly triggered by HX alone and 8% strain alone (*Figure 1B*). No significant synergy was found for the two stimulations: levels of collagen I are similarly elevated in all HX conditions regardless of strain applied. Correspondingly, collagen I staining exhibits similar intensity in all 8% strain conditions regardless of the oxygen stimulation employed. After only 12 hr of stimulation, collagen I fluorescence showed similar trends, with the exception of HX combined with 8% cyclic strain which provided a greater synergistic effect compared to the single stimulations (*Figure 1—figure supplement 1*). We also report that culturing CFs at NX levels does not induce significant changes in collagen I presence.

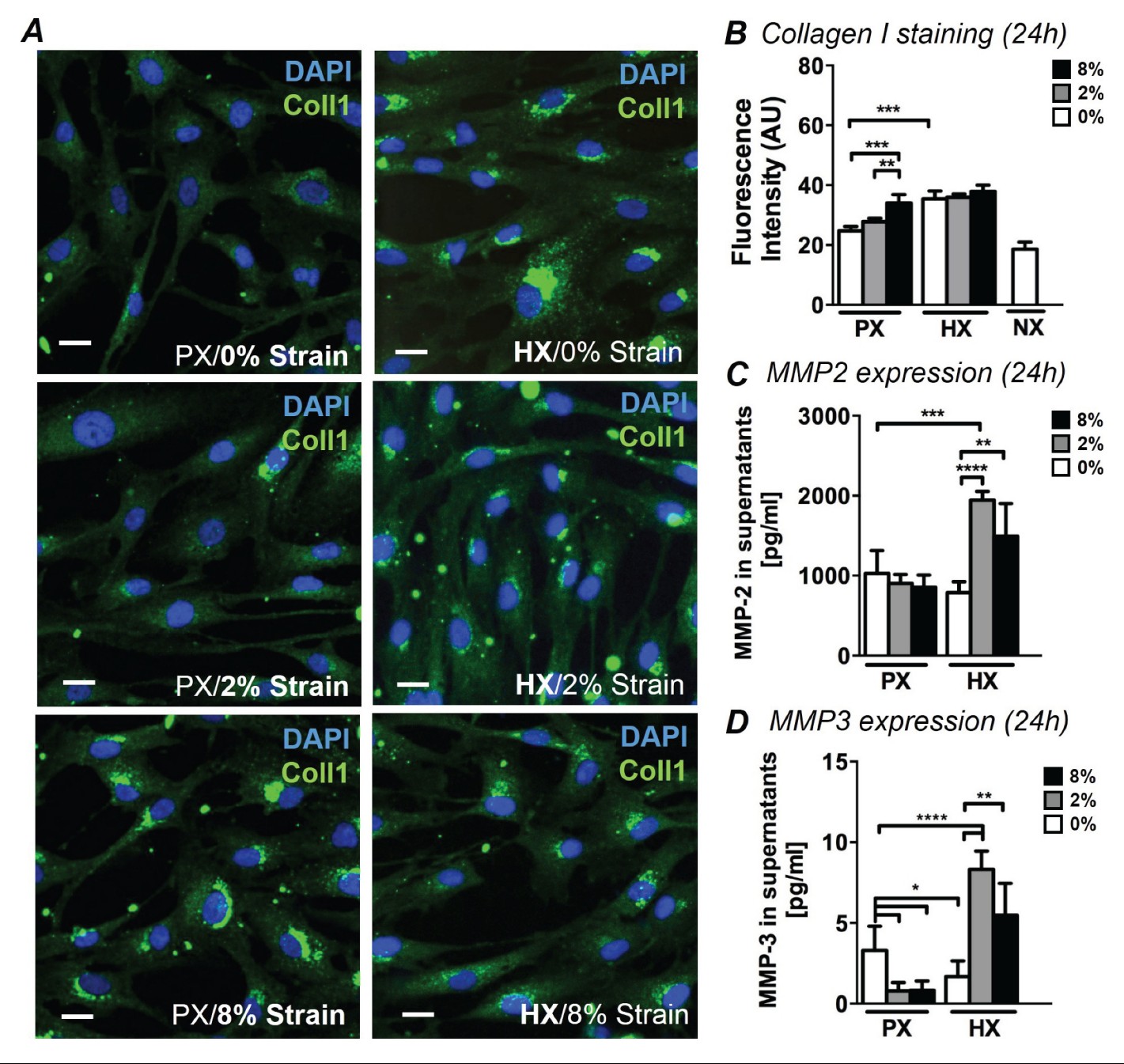

**Figure 1.** Analyses of early ECM adaptive responses by CFs subject to combined environmental stimulation. (A) Representative images of CFs fixed after 24 hr stimulations and stained for collagen I (green) and DAPI (blue). Scale bars = 20 μm. An intense perinuclear staining is observed in CFs stimulated either by 8% mechanical strain or by HX. (B) Quantitative fluorescence intensity analyses on Collagen I staining plotted as graph. Collagen I staining is increased by HX, 8% strain or NX conditions. Data collected from one cell donor, two independent experiments, minimum number of experimental replicates n = 4, technical replicates (multiple images per replicate) n = 3. Detected amounts of MMP-2 (C) and MMP-3 (D) in supernatants of CFs. Both MMP-2 and MMP-3 expression is significantly increased in combined HX and mechanical strain stimulation. Protein secretion data collected from one cell donor, two independent experiments, minimum number of experimental replicates n = 4, technical replicates n = 2. White histograms correspond to 0% strain conditions, grey histograms to 2% strain conditions and black histograms to 8% strain conditions. Two-way ANOVA test was performed for all groups. *p<0.05, **p<0.01, ***p<0.001, ****p<0.0001. One outlier measurement in MMP-3 expression was detected by performing Grubb's test (α = 0.05; p<0.05) and removed from the analysis.

The following source data and figure supplement are available for figure 1:

*Figure 1 continued on next page*

*Figure 1 continued*

**Source data 1.** Collagen intensity levels.
**Figure supplement 1.** Quantitative fluorescence intensity analyses on collagen I staining performed on CFs fixed after 12 hr of combined stimulation.

The expression of MMP-2 and MMP-3 in CFs supernatants (*Figure 1CD*) was significantly influenced by combined mechanical strain and oxygen changes. MMP-2 was more prominently expressed than MMP-3 (>100X higher). Nevertheless, the expression of both enzymes was similarly regulated by the pattern of stimulations applied: a significant two-fold increase in MMP-2 and MMP-3 expression was induced by the combination of HX and mechanical strain, with a more pronounced increase observed in the HX/2% strain combined conditions. MMP-3 expression was negatively affected when CFs were subjected to mechanical strain at PX or HX stimulus alone.

## Cell proliferation

During the proliferative phase of myocardial healing, excessive CF proliferation is observed as well as a phenotypic switch giving rise to the fibrotic response. In order to understand how environmental stimuli modulate the proliferation of human CFs, we examined the fraction of mitotic cells (PHH3+/DAPI) under combined mechanical stimulation and changes in oxygen tension. *Figure 2A* shows representative images of CFs stained for PHH3 and DAPI. Without application of cyclic strain, we observed a two-fold significant increase in the number of mitotic cells (*Figure 2B*) when CFs were stimulated with HX compared to their PX counterparts. Mechanical strain significantly impacted cell proliferation within the PX culture condition: culturing CFs at PX and subjecting them to 2% strain induced a striking increase in mitotic cells compared to no strain control and 8% strain cultures. This strain intensity-dependent effect is in line with our previous investigations of CF proliferation under mechanical strain in NX conditions (*Ugolini et al., 2016*). After 12 hr of stimulation, the proliferative increase with HX is not statistically significant, while we report a significant synergistic decrease of mitotic cells induced by the combination of 8% strain and HX (*Figure 2—figure supplement 1*).

In addition, we performed mitotic cell counts on CFs cultured under NX conditions without mechanical strain in order to evaluate any influence of non-physiological oxygen environments on CF proliferation. Interestingly, CFs exhibited much greater mitosis under NX environments (approx. 5% mitotic cells) compared to PX environments (approx. 1% mitotic cells) denoting a strong proliferative effect of standard cell culture environments.

Further, we evaluated the effects of the environmental changes on YAP, a transcription factor mainly known as the principal effector of the Hippo proliferative pathway, a crucial pathway involved in cardiac regeneration and repair (*Papizan and Olson, 2014*; *Xin et al., 2013*). Recent literature has also described a parallel mechano-sensing associated role of YAP (*Dupont et al., 2011*; *Mosqueira et al., 2014*). YAP translocation from cytoplasm to nucleus leads to inactivation of Hippo pathway and a subsequent increase in cell proliferation. We therefore quantified nuclear localization of YAP by means of immunofluorescence, which revealed that the fraction of CFs displaying nuclear YAP is influenced by the combination of stimuli with trends similar to the fraction of mitotic cells (*Figure 2C,D*). In CFs cultured at PX, 2% strain induces higher YAP translocation into cell nuclei. CFs solely exposed to HX environment show a significant two-fold increase in nuclear YAP compared to a PX environment. Mechanical strain shows a negative interaction when combined with HX, with CFs subject to HX/8% strain condition showing lower fractions of nuclear YAP compared to CFs subject to HX alone. This effect can also be observed in the analysis of mitotic cells, although the result is not statistically significant. Interestingly, CFs cultured in an NX environment show a three-fold increase in nuclear YAP compared to PX conditions.

Finally, we detected the expression of PDGF in CF supernatants. PDGF has been described as a potent mitogen for CFs, is significantly over-expressed in in vivo models of heart injury (*Zhao et al., 2011*), and has recently been studied as a putative pharmacological target for attenuating adverse effects of cardiac fibrotic disease (*Liu et al., 2014*). Expression of PDGF in CF culture supernatant (*Figure 2E*) was significantly affected by HX stimulation: a four-fold increase in the detected amounts of PDGF was observed in HX compared to PX, without application of cyclic strain. Mechanical stimulation showed a negative interaction with the HX-induced secretion of PDGF: while levels of PDGF

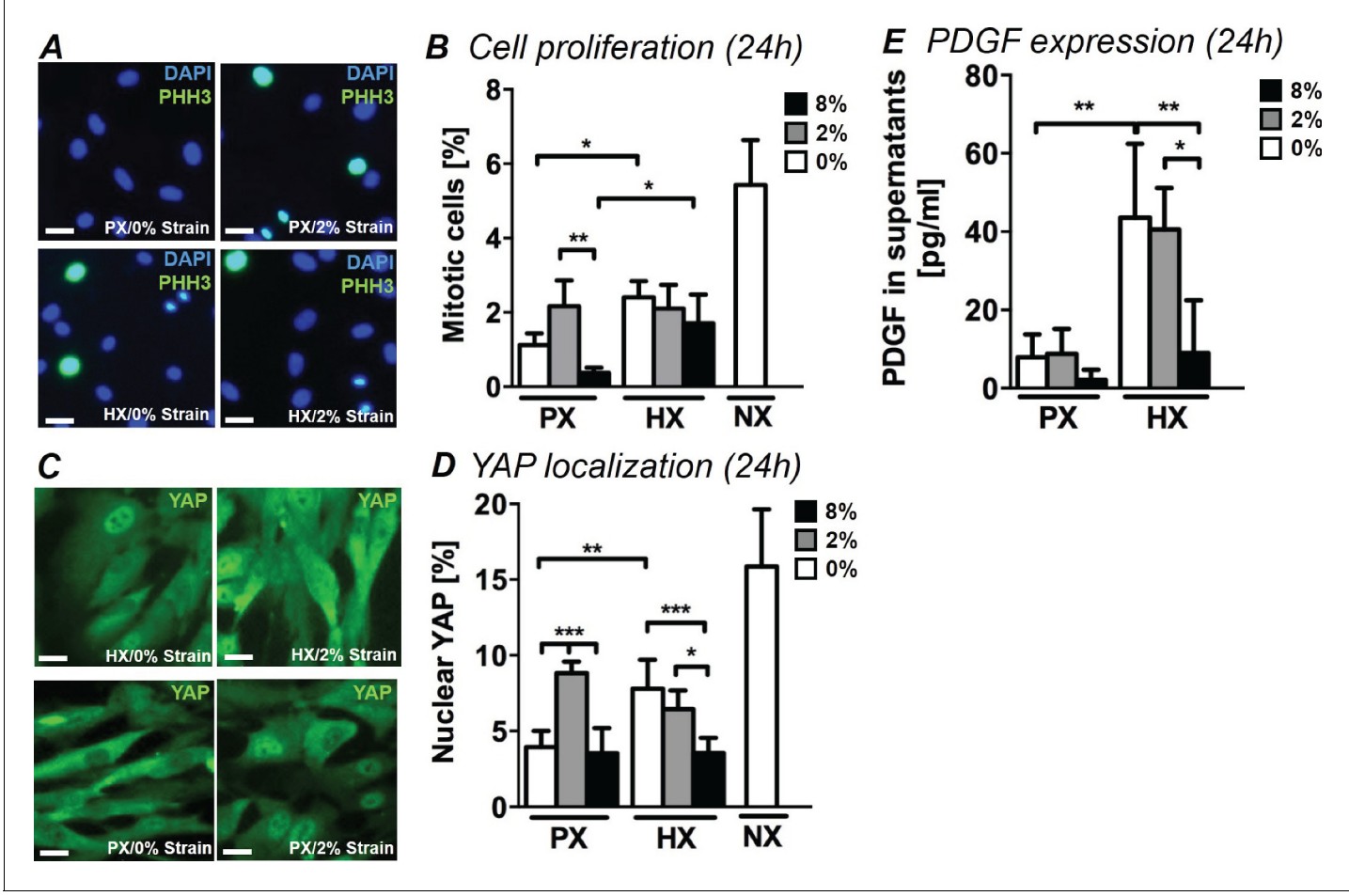

**Figure 2.** Analyses of CFs proliferation under combined environmental stimulation. (**A**) Representative images of CFs fixed after 24 hr stimulations and stained for PHH3 (green) and DAPI (blue). Scale bars = 50 μm. (**B**) Quantitative analyses of mitotic cell fraction plotted as graph. Cell mitosis is increased by either HX alone and by 2% strain at PX. (**C**) Representative images of CFs fixed after 24 hr stimulations and stained for YAP (green) and DAPI (blue). Scale bars = 50 μm. (**D**) Quantitative analyses of cells presenting nuclear YAP plotted as graph. YAP translocation into nuclei is increased by either HX or by 2% strain at PX. Data in Panels B and D were collected from one cell donor, two independent experiments, minimum number of experimental replicates n = 4, technical replicates (multiple images per replicate) n = 3. (**E**) Detected amounts of PDGF in supernatants of CFs plotted as graph. PDGF expression is significantly increased in HX alone and HX combined with mechanical strain stimulation (2% strain). Protein secretion data collected from one cell donor, two independent experiments, minimum number of experimental replicates n = 4, technical replicates n = 2. White histograms correspond to 0% strain conditions, grey histograms to 2% strain conditions and black histograms to 8% strain conditions. Two-way ANOVA test was performed for mitotic cells and nuclear YAP, whereas Kruskal-Wallis test was performed on PDGF expression. *$p<0.05$, **$p<0.01$, ***$p<0.001$.

The following source data and figure supplement are available for figure 2:

**Source data 1.** Mitotic cells and YAP localization data.

**Figure supplement 1.** Quantification of mitotic cells (PHH3[+]/DAPI) performed on CFs fixed after 12 hr of combined stimulation (PX, HX and mechanical strain).

expression remain elevated under HX/2% strain, application of 8% cyclic strain shows a negative regulation of PDGF secretion back to PX levels of expression.

## Cytokine secretion

Acute inflammatory tissue responses after myocardial injuries are thought to be largely based on cellular signaling by means of secreted factors. The precise contribution of CFs in the regulation of the early inflammatory response in myocardial remodeling is still to be determined, however, recent

reports suggest that CFs play a key role in modulating a functional inflammasome that includes inflammatory and pro-fibrotic cytokines (*Kawaguchi et al., 2011*; *Lindner et al., 2014*; *Turner, 2016*). We therefore studied the amount of inflammatory cytokines (namely interleukins IL-1$\beta$, IL-6 and TNF-$\alpha$) and of pro-fibrotic TGF-$\beta$ in CF culture supernatants after 24 hr of combined stimulations (*Figure 3*).

TGF-$\beta$ is a major contributor to fibrotic responses, widely studied across a variety of tissues and organs. During post-injury myocardial remodeling, TGF-$\beta$ is known to stimulate a wide range of CF responses from phenotypic switch to collagen synthesis and ECM remodeling (*Leask, 2007*). CFs stimulated with a combination of HX and mechanical strain (specifically 2% strain) were found to increase secretion of TGF-$\beta$ by approximately two-fold. CFs cultured at PX expressed lower amounts of TGF-$\beta$, regardless of the mechanical strain applied (*Figure 3A*).

IL1-$\beta$ is one of the first cytokines detected in vivo after myocardial injury (*Guillén et al., 1995*) and is known to stimulate CF response including the production of MMPs (*Brown et al., 2007*; *Guo et al., 2008*; *Siwik et al., 2000*). Interestingly, we found that a significant two-fold increase in IL1-$\beta$ expression occurred only when CFs were subjected to the combination of HX and mechanical strain, particularly with 2% strain (*Figure 3B*).

Conversely, we observed how the expression of inflammatory cytokines TNF-$\alpha$ and IL-6 was not influenced by the application of environmental stimuli. We detected extremely low amounts of TNF-$\alpha$ (*Figure 3C*), with non-significant variations across all stimulation conditions. IL-6 (*Figure 3D*) was abundantly expressed by CFs yet none of the differences in its expression were found to be statistically significant.

## Myofibroblast differentiation

In the remodeling myocardium, CFs undergo a phenotypic modulation to myofibroblasts, a motile and contractile cell type that maintains tissue integrity and promotes scar formation and tissue fibrosis (*Santiago et al., 2010*). The hallmark of this phenotype switch is the expression of alpha-smooth muscle actin ($\alpha$SMA) that, by being incorporated into actin stress fibers, confers increased mechanical and motile capabilities to differentiated myofibroblasts (*Baum and Duffy, 2011*). To analyze whether CFs differentiation into myofibroblasts took place under environmental changes we evaluated expression and localization of $\alpha$SMA after 24 hr of combined stimulation. Immunostainings of CFs (*Figure 4*) reveal that a basal expression of cytoplasmic $\alpha$SMA is present in all combined stimulation conditions. However, under no experimental condition did CFs exhibit co-localization of $\alpha$SMA (red) and actin stress fibers (green), indicating that differentiation into myofibroblasts was not induced by our pattern of environmental stimuli. Conversely, stimulating CFs by supplementing TGF-$\beta$ in the culture medium (a known inducer of CF differentiation to myofibroblasts) caused the appearance of cells exhibiting superimposed $\alpha$SMA and actin signal (yellow).

## Discussion

Microenvironmental changes taking place in the injured myocardium trigger tissue responses that are orchestrated between multiple cell types (mainly cardiac myocytes, cardiac fibroblasts and immune cells) and involve multiple cell functions and signaling pathways. The outcome of this process is a substantial remodeling of the injured myocardial region by ECM-producing CFs that often results in excessive and dysfunctional alterations of the myocardial architecture and functionality. The elucidation of pathological cellular routes leading to tissue-level failure can clearly benefit from effective in vitro studies capable of dissecting and analyzing responses to multiple physiological and pathological conditions. Microfluidic technologies have recently enabled the development of advanced cardiac models by integrating key environmental cues in cardiac cell cultures (*Agarwal et al., 2013*; *Marsano et al., 2016*; *Pavesi et al., 2015*; *Uzel et al., 2014*). We here investigated CF behaviors in a compact multi-chamber in vitro platform designed to perform cell cultures under combined mechanical stimulation and changes in oxygen levels. This novel tool enabled us to perform for the first time a systematic evaluation of early CF responses to the combination of oxygen levels and mechanical strain regimes that facilitate a more realistic model of cardiac physiology and pathology.

Despite being overlooked in most previous studies, rigorous replication of in vivo oxygen tensions proved fundamental for interpreting physiological and pathological CF responses (*Roy et al., 2003*). Two oxygen conditions were tested: physiological oxygen levels experienced by CFs in the

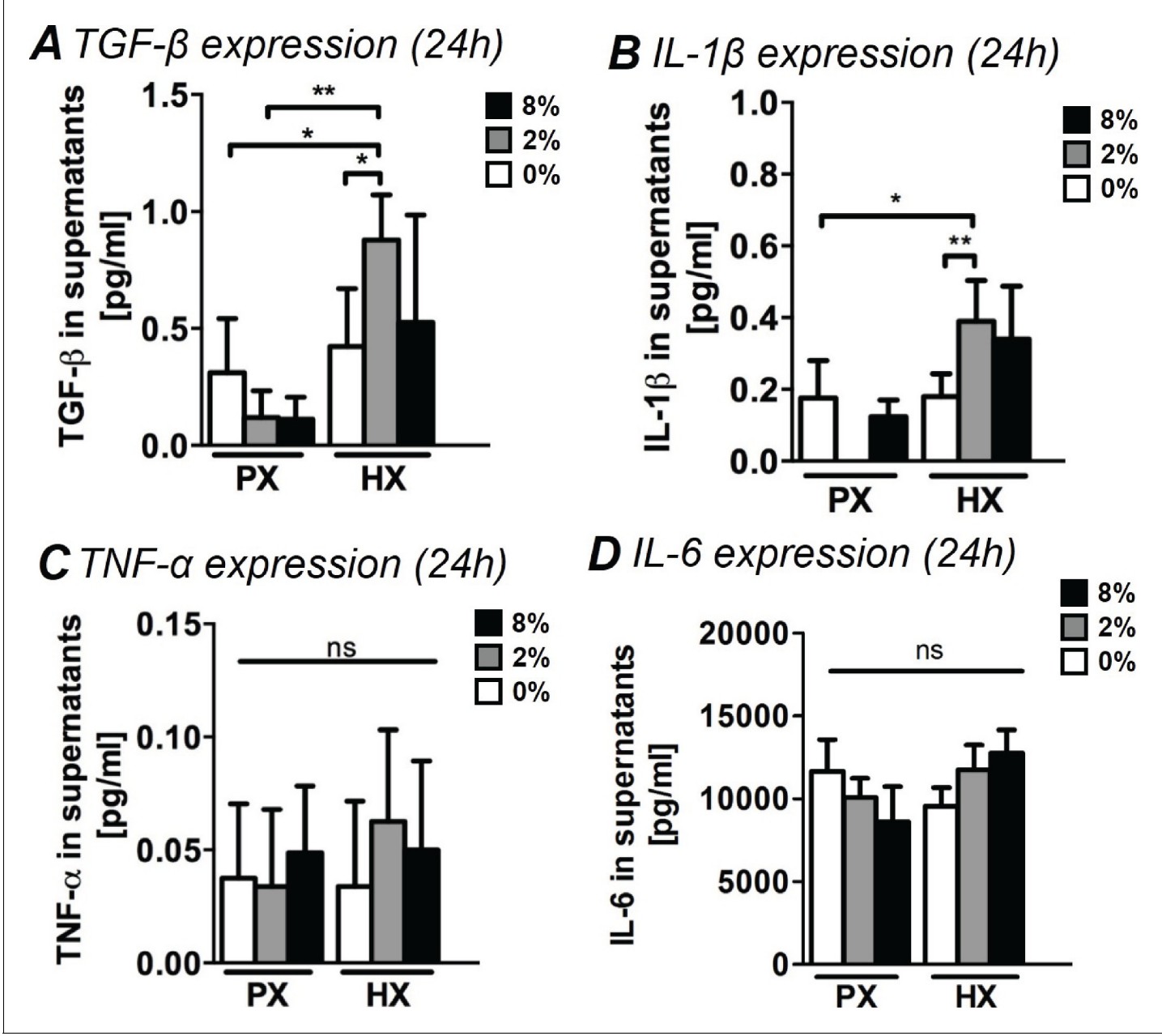

**Figure 3.** Analyses of pro-fibrotic and pro-inflammatory cytokines in supernatants of CFs. (A) TGF-β expression in supernatants. TGF-β expression resulted significantly increased by the combination of HX and mechanical strain (2% strain). (B) Expression of IL-1β in CFs supernatants. IL-1β expression resulted significantly upregulated by the combination of HX and mechanical strain (2% strain). (C) TNF-α expression in supernatants. Extremely low amounts were detected and environmental stimulations do not induce significant differences in the expression. (D) IL-6 expression in supernatants. IL-6 resulted abundantly expressed by CFs with no statistically significant differences induced by environmental stimulations. Protein secretion data from each marker was collected from one cell donor, two independent experiments, minimum number of experimental replicates n = 4, technical replicates n = 2. White histograms correspond to 0% strain conditions, grey histograms to 2% strain conditions and black histograms to 8% strain conditions. Kruskal-Wallis tests were performed for all groups except for IL-6 analyzed with Two-way ANOVA tests. *p<0.05, **p<0.01, ns = non-significant.

The following source data is available for figure 3:

**Source data 1.** Protein expression data.

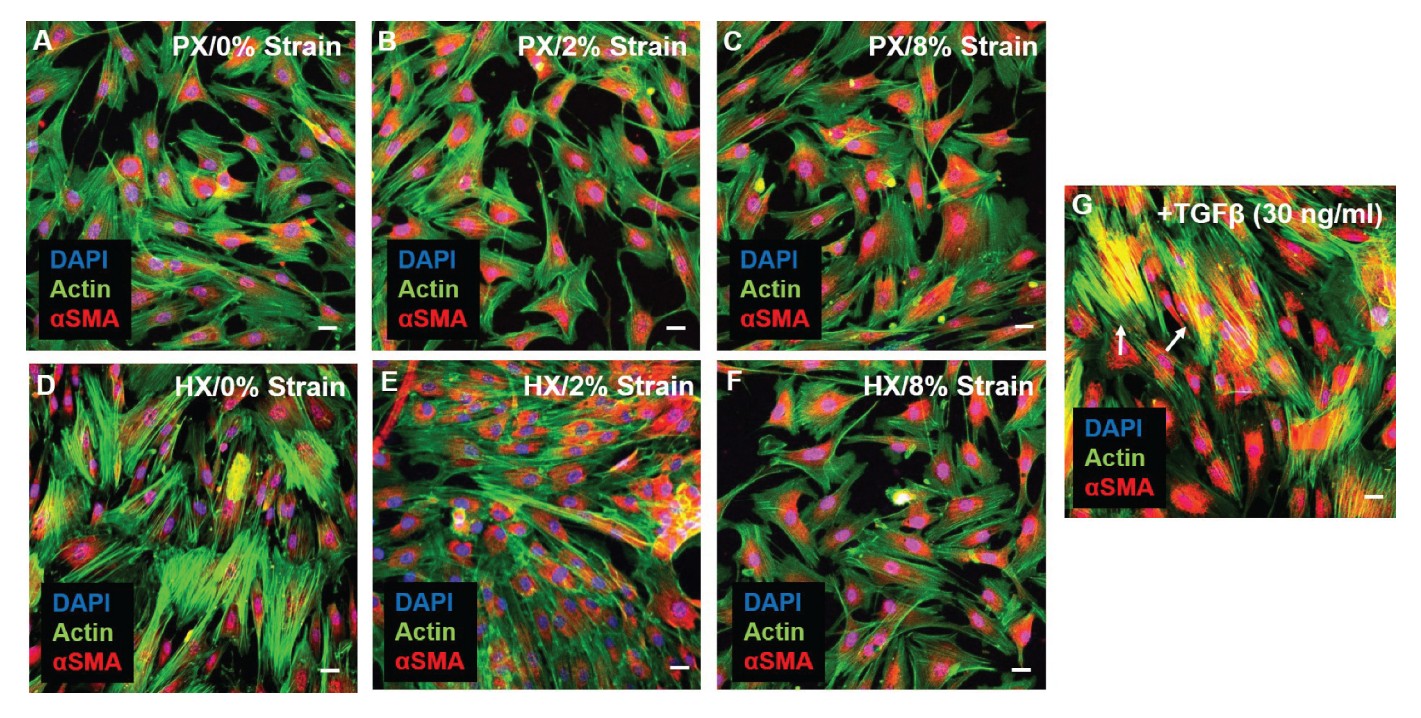

**Figure 4.** Representative images of CFs fixed after 24 hr combined stimulations (**A–F**) and stained for actin (green) and αSMA (red), nuclei were counterstained with DAPI (blue). Scale bars = 20 µm. αSMA is similarly expressed in all conditions and diffused in perinuclear and cytoplasmic localization. (**G**) Positive control for αSMA immunofluorescence through stimulation with 30 ng/ml TGF-*β*: CFs exhibit co-localization of αSMA staining with actin stress fibers (resulting in yellow signal) denoting differentiation into myofibroblasts (white arrows). Representative images were collected from a pool of images obtained from one cell donor, one independent experiment, minimum number of experimental replicates n = 4, technical replicates n = 3.

myocardium (5% $O_2$, PX) and a rapid decrease of oxygen levels resembling an acute ischemic event (1% $O_2$, HX). We also performed analyses on CFs cultured under non-physiological normoxic (NX) conditions and confirmed a substantial impact of NX compared to PX, especially on cell proliferation.

The mechanical microenvironment is known to change upon cardiac injury. It is known that injured myocardial regions are characterized by an early reduction of mechanical load due to loss of contractile cardiac myocytes (*Eek et al., 2010*; *Pfeffer and Braunwald, 1990*; *Picard et al., 1990*; *Tennant and Wiggers, 1935*). We therefore focused on evaluating how significant changes in mechanical strain affect cellular responses. To explore these effects we employed low cyclic strain mimicking reduced contractility (2% strain), physiological cyclic strain (8% strain) and a static control. These values were based on cardiac imaging studies performed shortly after myocardial injury (*Dandel et al., 2009*; *Vartdal et al., 2007*) and were previously reported to elicit strain-dependent effects in CFs (*Ugolini et al., 2016*).

Pathological remodeling of the myocardium involves activation of multiple CF cell functions. In a significant step forward compared to previous studies, our analyses encompassed most known pathological responses of CFs: production of ECM proteins and ECM-remodeling enzymes, cellular proliferation, secreted pro-fibrotic or inflammatory cytokines and myofibroblast differentiation. This in-depth assessment demonstrates how environmental factors participate in driving CF pathological responses and elucidates the existence of separate and synergistic roles of mechanical stimulation and oxygen levels in directing cellular functions relevant to early phases of cardiac fibrotic remodeling.

Our findings demonstrated that upon sensing of hypoxic environments CFs quickly (<24 hr) activate proliferative responses (*Table 1*). We propose that the increase in cell proliferation is mediated by the Hippo pathway given the correlation between nuclear YAP translocation and mitotic cells

**Table 1.** Summary of the main cellular responses observed in experiments of combined environmental stimulation. Cellular responses were included in the corresponding stimulation condition if the measured parameter was significantly different from the parameter at reference condition PX(5% $O_2$)/0% strain (single arrow ↑) and if more than two-fold significant difference was observed compared to PX(5% $O_2$)/0% strain condition (double arrow ↑↑). Different background cell color are added as a function of the number of significant cellular responses observed (yellow: 2 or less changes; orange: up to 5 changes; red: more than 5 changes).

| | 0% Strain | 2% Strain | 8% Strain |
|---|---|---|---|
| HX (1% $O_2$) | ↑↑ *Mitosis*<br>↑↑ *Nuclear YAP*<br>↑↑ *PDGF*<br>↑ *Collagen*<br>↓ *MMP-3* | ↑ *Mitosis*<br>↑ *Nuclear YAP*<br>↑↑ *PDGF*<br>↑ *Collagen*<br>↑↑ *MMP-2–3*<br>↑↑*TGF-β*<br>↑ *IL-1β* | ↑ *Collagen*<br>↑ *MMP-2* |
| PX (5% $O_2$) | *Reference condition* | ↑ *Mitosis*<br>↑↑ *Nuclear YAP*<br>↓ *MMP-3* | ↑ *Collagen*<br>↓ *MMP-3* |

(PHH3[+]). YAP is widely recognized as a key factor in cardiac repair (*Papizan and Olson, 2014*; *Xin et al., 2013*). We found, however, that in contrast with other studies performed at NX (*Codelia et al., 2014*), YAP nuclear translocation is not related to increasing cyclic mechanical strain. In addition, an increased secretion of mitogenic PDGF by CFs at HX suggests a possible autocrine CF mechanism aimed at modulating proliferation upon pathological myocardial conditions.

We also showed that hypoxia alone causes an increased collagen presence in CFs cultures. This is consistent with data obtained in previous works focused on collagen production by CFs subject to HX (*Tamamori et al., 1997*). We therefore demonstrated how HX alone induces pro-fibrotic responses in CFs.

A significant novel insight observed in the present study is that the combination of HX and mechanical strain incrementally instructed further pathological cell responses. Indeed, the production of pro-fibrotic TGF-β, inflammatory cytokine IL1-β and matrix-degrading enzymes (MMP-2 and MMP-3) significantly increased only when cells were subject to combined HX and low levels of cyclic strain (HX/2% strain). TGF-β is markedly upregulated in vivo in regions of cardiac tissue injury and is known to act in vitro on CFs phenotypic switch, proliferation and collagen synthesis (*Leask, 2007*; *Lijnen et al., 2000*). IL1-β has been described in vivo as an early (<10 hr) post-injury inflammatory cytokine (*Guillén et al., 1995*) and CFs are thought to be one of the main cellular sources of IL1-β (*Turner, 2014*). MMPs are known to be primarily involved in the remodeling of the injured myocardium and it is worth noting that previous studies reported inhibitory effects of HX alone on MMP expression (*Riches et al., 2009*). In another study, mechanical stress did not alter MMP expression unless cells were cultured in serum deprivation (*Tyagi et al., 1998*). This context strengthens the hypothesis of a combined contribution of environmental conditions in driving MMP secretion by CFs, and the importance of reproducing complex environmental changes in vitro. In addition, literature suggests a direct effect of TGF-β and IL1-β on the expression of MMPs in CFs (*Brown et al., 2007*; *Stawowy et al., 2004*) and we observed how the expression levels of TGF-β and IL1-β under combined stimulation are in line with the altered expression of MMPs. Further studies may therefore be directed towards the elucidation of such possible autocrine mechanisms regulating CFs-mediated ECM remodeling. Notably, no experimental condition led to myofibroblast differentiation, with the exception of the positive control (additional biochemical stimulation with TGF-β), thus highlighting the importance of soluble pro-fibrotic factors for this aspect of CF pathological activation.

Intriguingly, we found that when cells were subject to physiological cyclic strain (8% strain) together with HX condition, most of the above pathological CF responses were not observed and we found only a modest increase in collagen presence. Similarly, when oxygen conditions resemble those of healthy tissue (PX), cyclic mechanical strain only results in increased collagen presence (at 8% strain) or in increased proliferation (at 2% strain). These results are in line with previous studies

(performed in NX environments) on mechanically stimulated CFs (*Carver et al., 1991*; *Husse et al., 2007*; *Ugolini et al., 2016*).

Overall, these outcomes suggest that CFs respond differently to different regimes of cyclic strain, to different oxygen conditions and to combinations of these. In the context of an acute myocardial insult, loss of contractility and decrease in sensed oxygen levels are able to instruct a consistent response of CFs that involves matrix remodeling, inflammatory and proliferative behaviors. Our combined conditions allowed us to report that a mechanical environment resembling full contractility (8% strain) may hold a protective potential and resulted in attenuating CFs inflammatory, remodeling and pro-fibrotic responses taking place in HX.

These findings help to validate an on-chip model capable of exploring in vitro early fibrotic responses elicited in the injured myocardium. To estimate how this novel approach quantitatively correlates with standard readouts associated with cardiac fibrotic responses, we compared the fold changes of CF markers assessed in our model to those observed in standard in vitro models. Standard approaches typically involve the biochemical stimulation of CFs with known concentrations of pro-fibrotic (e.g., TGF-$\beta$) or inflammatory (e.g., interleukins) molecules. We collected literature studies where protein expression of relevant markers was measured at early stages of CFs culture under pro-fibrotic/inflammatory conditions. *Table 2* shows quantitative comparisons between standard models and our on-chip model. Our model correlates well with quantitative readouts involving ECM, ECM-remodeling enzymes (collagen and MMPs expression) and proliferative effects, although contradictory studies also showed decreased CF proliferation under pro-fibrotic stimuli. Notably, MMP levels in serum were shown to increase by approximately two-fold in patients after 24 hr from acute myocardial injury diagnosis (*Morishita et al., 2015*). As for CF secretion of pro-fibrotic and inflammatory factors, our model was able to capture the triggering of PDGF expression by CFs, an aspect not yet described by standard in vitro approaches. Interestingly, this also correlated with in vivo increments in plasma levels of PDGF (approximately two-fold increment) at the early onset (12 hr) of myocardial injury in human patients (*Koizumi et al., 2015*). Additionally, our results correlated with secretion of TGF-$\beta$ and IL-1$\beta$, while our environment-based model did not induce significant changes to the secreted levels of IL-6 and TNF-$\alpha$ as in standard models.

This last missing readout suggests that indirect paracrine action provided by other cellular actors involved in myocardial homeostasis and typically modeled in standard biochemical stimulation approaches is likely to play a role in driving the inflammatory responses that are not captured in our model. For instance, immune cells such as monocytes or macrophages are known to infiltrate the injured myocardium, supporting the inflammatory environment and providing additional paracrine cytokines to CFs (*van Amerongen et al., 2007*; *Frantz et al., 2013*; *Nahrendorf et al., 2010*). In addition, cross talk between cardiac myocytes and CFs has been widely described (*Cartledge et al., 2015*; *Zhang et al., 2012*) and shown to drive CF behaviors such as myofibroblast differentiation (*Tsoporis et al., 2012*) and increased production of inflammatory molecules (*Bowers et al., 2010*). These indirect mechanisms could help evoke additional responses not captured in our model or could increase to an even greater magnitude the ones that we described as driven by environmental signals. Direct effects of cardiomyocyte contractility also mechanically influence CF behavior in vivo, however, this aspect is recreated in vitro by our system that exposes cultured CFs to uniform but tunable mechanical signals, thus proving advantageous for improving the clarity of readouts. Further improvements to the current model, such as the addition of paracrine signaling with other cell types, represents a promising future direction for this work, with the aim of developing a more accurate on-chip model combining fine control over the microenvironment with realistic biochemical signaling.

## Conclusions

In summary, we propose an innovative model that recreates a pathological environment for understanding CF responses during cardiac injury. Replicating for the first time the combination of oxygen changes and mechanical cues, we revealed how these environmental stimulations combine to trigger CF pathological responses and highlighted emerging adaptive cellular mechanisms. We found that mimicking the combination of HX and reduced contractility proved crucial in eliciting inflammatory and fibrotic remodeling responses of CFs, while individual environmental stimuli only regulated proliferative and collagen-related responses. These insights have impact on future studies of pathological myocardial remodeling and the in vitro model here described provides a tool for better understanding pathological mechanisms and tailoring reparative strategies.

**Table 2.** Summary of literature studies investigating early CFs responses under standard pro-fibrotic and inflammatory stimulation. Quantitative fold-changes of selected markers (fibrotic vs. control condition) are reported for studies that carried out protein expression assays or proliferation assays at early stimulation timepoints and compared to our on-chip model (grey background). The on-chip model well correlates to standard in vitro models in terms of ECM remodeling markers (Collagen 1, MMPs), cell proliferation and pro-fibrotic or inflammatory cytokynes (TGF-$\beta$ and IL-1$\beta$). The on-chip model did not elicit production of IL-6 and TNF-$\alpha$ as compared to standard in vitro models. Assays: WB=Western Blot; ELISA= Enzyme-linked immunosorbent assay; HP= Hydroxyproline assay; CCK-8= Cell-counting kit eight proliferation assay; FACS= Fluorescence-activated cell sorting; Count= Standard cell counting; BrdU= Bromodeoxyuridine proliferation assay. Chemical stimuli: Ang II= Angiotensin II; TGF-$\beta$= Transforming Growth Factor-$\beta$; IL-1$\alpha$/IL-1$\beta$= Interleukin-1$\alpha$/1$\beta$; TNF-$\alpha$= Tumor Necrosis Factor-$\alpha$. Source data file linked to **Table 2** shows this comparison data plotted in a chart.

### Collagen Expression

| Ref. | Species | Assay | Stimulus | Culture Time | Change vs. control (standard model) | Change vs. control (24 hr on-chip model) |
|---|---|---|---|---|---|---|
| (*Xiao et al., 2016*) | Mouse | WB | Ang II | 24 hr | ↑1.5-fold | ↑1.45-fold |
| (*Guo et al., 2016*) | Rat | WB | TGF-$\beta$ | 48 hr | ↑2.2-fold | |
| (*Pan et al., 2013*) | Rat | WB | TGF-$\beta$ | 24 hr | ↑1.5-fold | |
| (*Peng et al., 2010*) | Human | HP | TGF-$\beta$ | 48 hr | ↑1.6-fold | |
| (*Li et al., 2015*) | Rat | WB | Ang II | 24 hr | ↑2.1-fold | |

### MMP-2 Expression

| Ref. | Species | Assay | Stimulus | Culture Time | Change vs. control (standard model) | Change vs. control (24 hr on-chip model) |
|---|---|---|---|---|---|---|
| (*Xiao et al., 2016*) | Mouse | WB | Ang II | 24 hr | ↑1.8-fold | ↑2-fold |
| (*Rhaleb et al., 2013*) | Rat | WB | IL-1$\beta$ | 72 hr | ↑2-fold | |
| (*Brown et al., 2007*) | Rat | WB | IL-1 | 48 hr | ↑2.2-fold | |
| (*Li et al., 2015*) | Rat | WB | Ang II | 24 hr | ↑2-fold | |

### MMP-3 Expression

| Ref. | Species | Assay | Stimulus | Culture Time | Change vs. control (standard model) | Change vs. control (24 hr on-chip model) |
|---|---|---|---|---|---|---|
| (*van Nieuwenhoven et al., 2013*) | Human | WB | IL-1$\alpha$ | 24 hr | ↑2-fold | ↑2.7-fold |
| (*Brown et al., 2007*) | Rat | WB | IL-1 | 48 hr | ↑2-fold | |

### Cell proliferation

| Ref. | Species | Assay | Stimulus | Culture Time | Change vs. control (standard model) | Change vs. control (24 hr on-chip model) |
|---|---|---|---|---|---|---|
| (*Xiao et al., 2016*) | Mouse | CCK-8 | Ang II | 24 hr | ↑2-fold | ↑1.9-fold |
| (*Guo et al., 2016*) | Rat | FACS | TGF-$\beta$ | 48 hr | ↑2-fold | |
| (*Porter et al., 2004*) | Human | Count | TNF-$\alpha$ | 4 days | ↑1.5-fold | |
| (*Dobaczewski et al., 2010*) | Mouse | BrdU | TGF-$\beta$ | 24 hr | ↓2-fold | |
| (*Vivar et al., 2016*) | Rat | FACS | TGF-$\beta$ | 72 hr | ↓2-fold | |
| (*Ai et al., 2015*) | Rat | CCK-8 | Ang II | 48 hr | ↑1.8-fold | |
| (*Li et al., 2015*) | Rat | MTT | Ang II | 24 hr | ↑2-fold | |

### TGF-$\beta$ Expression

| Ref. | Species | Assay | Stimulus | Culture Time | Change vs. control (standard model) | Change vs. control (24 hr on-chip model) |
|---|---|---|---|---|---|---|
| (*Gu et al., 2012*) | Mouse | WB | Ang II | 24 hr | ↑2-fold | ↑2.7-fold |
| (*Xiao et al., 2016*) | Mouse | WB | Ang II | 24 hr | ↑1.8-fold | |
| (*Ai et al., 2015*) | Rat | ELISA | Ang II | 48 hr | ↑2.5-fold | |
| (*Li et al., 2015*) | Rat | WB | Ang II | 24 hr | ↑1.9-fold | |

*Table 2 continued on next page*

| | | | | | IL-1β Expression | |
|---|---|---|---|---|---|---|
| Ref. | Species | Assay | Stimulus | Culture Time | Change vs. control (standard model) | Change vs. control (24 hr on-chip model) |
| (*Turner et al., 2009*) | Human | ELISA | IL-1α | 24 hr | ↑8-fold | ↑2.2-fold |
| (*Turner et al., 2009*) | Human | ELISA | TNF-α | 24 hr | ↑2-fold | |
| | | | | | TNF-α Expression | |
| Ref. | Species | Assay | Stimulus | Culture Time | Change vs. control (standard model) | Change vs. control (24 hr on-chip model) |
| (*Turner et al., 2009*) | Human | ELISA | IL-1α | 24 hr | ↑4-fold | ns |
| (*Humeres et al., 2016*) | Rat | Luminex | TGF-β | 72 hr | ↓4-fold | |
| (*Yokoyama et al., 1999*) | Rat | ELISA | Ang II | 6h | ↑5-fold | |
| | | | | | IL-6 Expression | |
| Ref. | Species | Assay | Stimulus | Culture Time | Change vs. control (standard model) | Change vs. control (24 hr on-chip model) |
| (*Turner et al., 2009*) | Human | ELISA | IL-1α | 24 hr | ↑19-fold | ns |
| (*Turner et al., 2009*) | Human | ELISA | TNF-α | 24 hr | ↑3.5-fold | |
| (*Turner et al., 2007*) | Human | ELISA | TNF-α | 24 hr | ↑2.8-fold | |

**Source data 1.** Data from *Table 2* plotted as chart.

## Materials and methods

### Experimental design

*Figure 5* shows a detailed representation of oxygen dynamics and mechanical strain conditions employed (*Figure 5A and B*) together with a timeline of experiments (*Figure 5C*). After cells seeding, CFs were kept in static incubation for 12 hr in order to allow for cell adhesion. Subsequently – being this initial time-point referred to as t0 – CFs seeded in the inlet and outlet wells of the culture chambers were manually scraped, aspirated and stimulations were started. We employed two mechanical stimulation regimes (*Figure 5B*): 2% strain or 8% strain at 1 Hz. Concerning the oxygenation stimulus applied, the microdevices were employed at a base environmental oxygen level corresponding to physoxia (5% $O_2$) in order to precisely model the oxygen levels in healthy myocardium until t0. Then, two oxygen concentration dynamics were imposed (*Figure 5C*): (i) a static incubation at 5% $O_2$ (physoxia, PX) for 24 hr, (ii) an abrupt reduction of oxygen concentrations to 1% $O_2$ that was maintained for 24 hr (hypoxia, HX). CFs were subject to all possible combinations of the above stimuli (six total conditions) for a total of 24 hr. Immunofluorescence analyses were performed after 12 hr and 24 hr of stimulation, while cell culture supernatants were collected and analyzed after 24 hr of stimulation. Immunofluorescence analyses were also performed on CFs cultured at NX levels under 0% strain condition to investigate responses induced by culturing CFs in non-physiological NX environments, a standard condition for most previous studies on CFs.

### Microdevice outline

The microdevice was fabricated as previously described (*Ugolini et al., 2016*). Briefly, four stretching units are arranged in a single device composed of a thin polydimethylsiloxane (PDMS) membrane sandwiched between two microstructured PDMS layers. *Figure 5—figure supplement 1* shows a picture of the microdevice together with a sketched cross-section of a stretching unit of the device. In each stretching unit, a central culture chamber (blue) is flanked by two actuation chambers (red) connected to a single actuation line, meant for vacuum application and straining of the central membrane. A lower fluidic channel (green), common to all units, is designed for environmental conditioning and flows below each central culture chamber. *Figure 5—figure supplement 1* also shows the stimulation system: microdevices were actuated by a vacuum line for generating mechanical strain and a gas line for controlling oxygen concentrations. As for the vacuum line, programmable

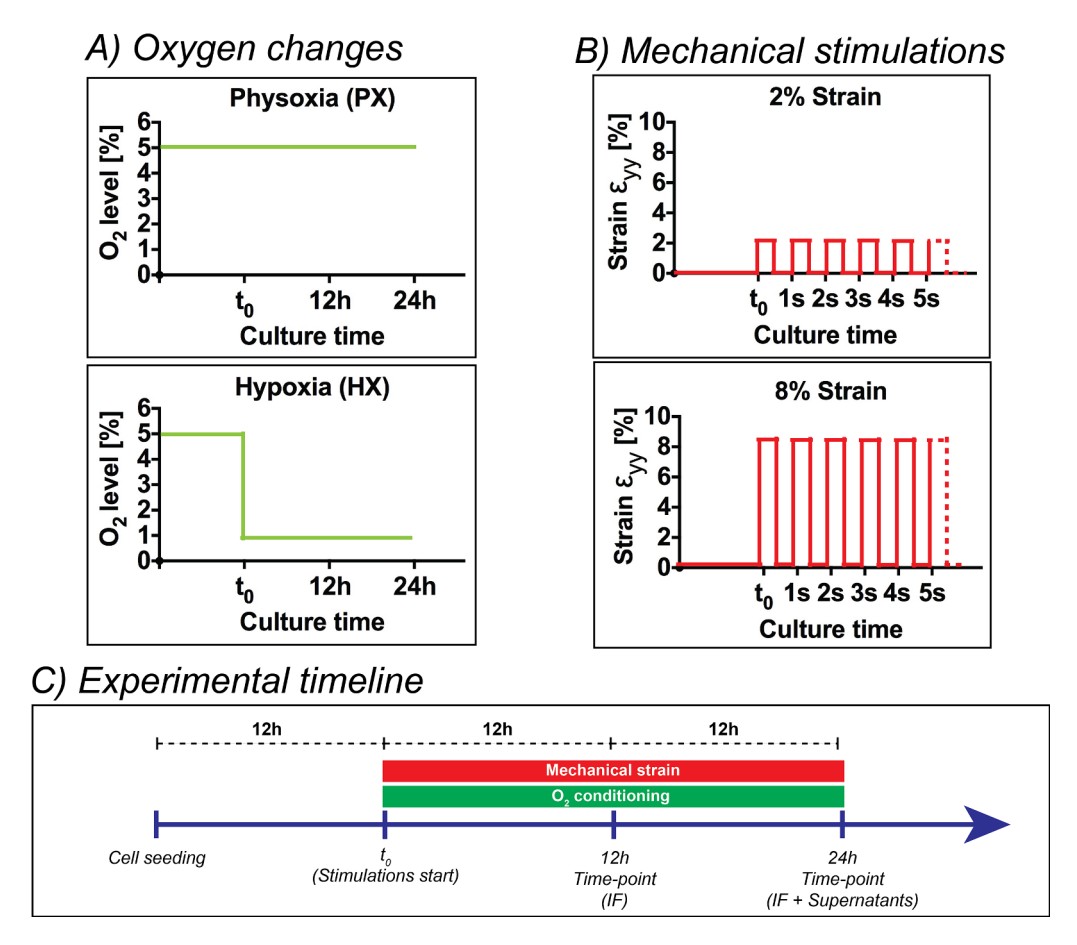

**Figure 5.** Overview of the experimental design of the present work. (A) Oxygen changes reproduced in the present work: an incubation at PX; an abrupt reduction to HX environments (1% $O_2$) maintained for 24 hr. (B) Mechanical stimulation regimes investigated in the present work: a static control at constant 0% strain; a cyclic mechanical strain stimulation at 2% strain and 1 Hz frequency; a cyclic mechanical strain stimulation at 8% strain and 1 Hz frequency. (C) Experimental timeline of the experiments performed: cells were seeded on microdevices and allowed to adhere to the culture membranes for 12 hr at PX and 0% strain before starting the combination of environmental stimuli. After 12 hr from the beginning of stimulations, we fixed samples for immunofluorescence analyses and after 24 hr we fixed samples for immunofluorescence analyses and collected supernatants for protein expression quantifications.

The following source data and figure supplements are available for figure 5:

**Figure supplement 1.** Overview of the microfluidic platform employed in the present work together with control system.

**Figure supplement 2.** Numerical modeling and experimental characterization of oxygen conditioning system included in the microdevices employed for the present work.

*Figure 5 continued on next page*

electromechanical valves modulated the vacuum to switch cyclically from atmospheric pressure to the desired vacuum pressure (namely −200 mmHg for 2% strain and −600 mmHg for 8% strain, applied for half a cycle at a rate of 1 Hz). The pneumatic stretch system was previously described and validated (*Ugolini et al., 2016*). The flowrate of humidified gas mixture (95% $N_2$; 5% $CO_2$) was regulated with a flowmeter and delivered to the microdevices through low-permeability gas tubing. Numerical modeling and experimental characterization of the oxygen control system is described in *Figure 5—figure supplement 2*. Numerical models of a 2D cross-section of the microdevice culture chamber show that the spatial distribution of oxygen tension is uniform along the width of the culture membrane, with limited border effect near the side walls (*Figure 5—figure supplement 2A,B*).

We employed a needle-based, fine-tip (spatial resolution ≈ 50 µm) oxygen sensor adjusted with a micromanipulator (PreSens, Germany) to reach the culture membrane. Measured oxygen concentration values confirm that the culture membrane is conditioned to ≈ 1% $O_2$ at varying flowrates and consistently throughout all four culture chambers (*Figure 5—figure supplement 2C,D*). The low-oxygen conditioning occurs within seconds, while reoxygenation occurs within minutes (*Figure 5—figure supplement 2E*). Membrane strains upon application of varying gas flowrates were measured as previously described (*Ugolini et al., 2016*). Gas flowrates of 5 ml/min and 15 ml/min do not cause significant membrane strain (*Figure 5—figure supplement 2F*).

## Cell culture

Normal human ventricular cardiac fibroblasts from one donor were purchased from Lonza (Lonza Bioscience, Singapore). Cells were cultured in FGM-3 medium (Lonza Bioscience, Singapore) and in a humidified incubator at 90% $N_2$, 5% $O_2$, 5% $CO_2$ at all times unless otherwise indicated. Microdevices were autoclave-sterilized, plasma-treated and coated with human fibronectin (Sigma-Aldrich, Singapore) for 30 min at room temperature. Cells were seeded for experiments at a passage number of four. After pre-loading each culture chamber of the microdevices with 80 µl of medium, 20 µl of cell suspension ($10^6$ cells/ml) were manually injected in the wells of the culture chambers. During experiments, microdevices were kept in humidified chambers to avoid culture medium evaporation.

## Immunofluorescence

Cells were fixed with 4% paraformaldehyde in PBS for 10 min. After 15 min of permeabilization with PBS containing 0.5% Triton-X, cells were blocked for 1 hr at room temperature with 3% bovine serum albumin. Cells were then probed overnight at 4°C with the following primary antibodies: anti-collagen I (mouse, AbCam, UK; RRID:AB_305411), to identify alterations in collagen I in the culture; anti-phospho-Histone-H3 (PHH3, Ser10, rabbit, Santa-Cruz, US; RRID:AB_2233067), for mitotic cells; anti-YAP (rabbit, Santa-Cruz, US; RRID:AB_2273277) to localize nuclear or cytoplasmic localization of the YAP/TAZ complex, a mechano-sensing associated transcription factor and main effector of the Hippo proliferative pathway; anti-αSMA (Smooth Muscle Actin, rabbit, AbCam, UK; RRID:AB_2223021) to identify myofibroblast differentiation from expression and localization of αSMA. CFs cultured under standard culture medium supplemented with 30 ng/ml TGF-$\beta$ (Sigma-Aldrich, Singapore) were considered positive controls for αSMA stainings. The following secondary antibodies were used for 2 hr at room temperature: goat anti-mouse Alexa Fluor 488 and goat anti-rabbit Alexa Fluor 564 (AbCam, UK). Nuclear staining was performed by incubating cells with DAPI. Negative controls were present for all immunofluorescence stainings.

## Image processing

Images were acquired with a Zeiss 710 Confocal microscope. Imaging parameters were not changed during acquisitions. For quantitative analyses of immunofluorescence markers, three images per culture chamber were taken at 10X magnification, thus sampling roughly half of the total area of the culture membrane and screening about 500 cells per replicate. Images were acquired from the central region of the culture membrane. Collagen I analyses were performed by intracellular fluorescence intensity quantification: the intracellular integrated density parameter was calculated by manually drawing outlines of about 50 cells per replicate from images, correcting for background intensity and normalizing for cell area. Cell proliferation analyses were performed by manually counting nuclei positive for PHH3 and dividing by the total number of nuclei (automatically counted) to estimate the fraction of mitotic cells. YAP localization analyses were performed by manually discriminating cytoplasmic or nuclear staining as previously described (*Codelia et al., 2014*; *Dupont et al., 2011*) and dividing by the total number of nuclei (automatically counted) to estimate the level of YAP nuclear translocation.

## Supernatant analyses

A volume of culture medium (100 µl) was collected per each culture chamber from the microfluidic devices after 24 hr. Supernatants were stored at −80°C prior to analysis. Residual cells and debris in the supernatant were removed by centrifugation. Concentrations of secreted factors in the

supernatants were assessed via multiplex bead-based array according to the manufacturer's instructions (Luminex, Austin, TX). Each measurement was run in duplicate.

## Statistical analyses

All data are presented as mean ± SD. Statistical comparisons were performed using GraphPad (Prism) software. All data were initially analyzed for normality using Kolmogorov-Smirnov tests. Two-way ANOVA tests followed by Bonferroni post-hoc tests were applied to determine statistical significance of differences and evaluate synergistic or separate contribution of mechanical strain stimulation and oxygen dynamics stimulation. When data groups did not pass normality tests, non-parametric statistical tests were employed (Kruskal-Wallis test). A p-value lower than 0.05 was considered significant. All data were collected from a pool of six independent experiments with a minimum number of biological replicates of four. We designed the approximate sample size required for the study by performing power analysis based on previously reported data on human cardiac fibroblasts (CF) proliferative responses under mechanical strain (*Ugolini et al., 2016*). In addition, we considered preliminary data obtained from human CFs cultured at different oxygen levels. An effect size was computed from this data (fraction of mitotic PHH3$^+$ cells under 2% vs. 8% strain: 2.0 ± 0.5 vs. 0.5 ± 0.3; expression of MMPs at physoxia vs. hypoxia: 1.02 ± 0.2 vs. 2.00 ± 0.1 ng/ml). An *a priori* power analysis performed with GPower (v.3.1) software assured that a sample size of n = 4 is sufficient to achieve $\alpha = 0.01$ and 1-$\beta$ (power) = 0.90.

## Acknowledgements

GSU was awarded a PhD Scholarship from the Italian Ministry of Education, supported by funds from InterPolytechnic Doctoral School mobility scholarship and by Progetto Rocca MIT-Italy exchange program. The research was supported by the National Research Foundation (NRF), Prime Minister's Office, Singapore, under its CREATE programme, Singapore-MIT Alliance for Research and Technology (SMART) BioSystems and Micromechanics (BioSyM) IRG.

## Additional information

### Funding

| Funder | Author |
|---|---|
| Singapore-MIT Alliance for Research and Technology Centre | Roger Kamm |
| National Research Foundation | Roger Kamm |

No external funding was received for this work.

### Author contributions

GSU, Conceptualization, Investigation, Methodology, Writing—original draft, Writing—review and editing; AP, Supervision, Investigation, Methodology, Writing—review and editing; MR, GBF, Supervision, Methodology, Writing—review and editing; RK, Supervision, Funding acquisition, Project administration, Writing—review and editing; MS, Conceptualization, Supervision, Methodology, Funding acquisition, Project administration, Writing—review and editing

### Author ORCIDs

Giovanni Stefano Ugolini, http://orcid.org/0000-0003-4775-6676
Andrea Pavesi, http://orcid.org/0000-0003-2777-1043
Monica Soncini, http://orcid.org/0000-0001-8607-7196

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
