## [Decision Letter]

Thank you for submitting your article "Human cardiac fibroblasts adaptive responses to controlled combined mechanical strain and oxygen changes" for consideration by *eLife*. Your article has been favorably evaluated by Fiona Watt (Senior Editor) and two reviewers, one of whom, Gordana Vunjak-Novakovic (Reviewer #1), is a member of our Board of Reviewing Editors.

The reviewers have discussed the reviews with one another and the Reviewing Editor has drafted this decision to help you prepare a revised submission.

The authors model the formation of fibrotic scar in human myocardium, using a simple microfluidic device to subject human cardiac fibroblasts to controllable levels of oxygen and mechanical strain. The central idea is to investigate the combinations of these two important signals that correspond to the healthy tissue, ischemic tissue and reoxygenated myocardium. Three known determining features of cardiac fibrosis were evaluated: remodeling of the extracellular matrix (collagen I, MMPs), proliferation of cardiac fibroblasts (mitotic cells, PDGF), and secretion of cytokines (inflammatory, pro-fibrotic). The main finding is that the combination of hypoxia (1% oxygen) and reduced contractility (2% strain) induced fibrotic responses in cardiac fibroblasts: production of MMPs and secretion of inflammatory and pro-fibrotic cytokines.

Summary:

The reviewers find that the paper describes an interesting in vitro model of cardiac fibrosis, by studying the responses of cardiac fibroblasts to combinations of mechanical strain and oxygen levels. The main criticism is that the study lacks mechanical insights.

Essential revisions:

The reviewers raise a number of concerns that must be adequately addressed before the paper can be further considered for publication in *eLife*. Some of the required revisions will likely require further experimentation within the framework of the presented studies and techniques.

1) The validation of the model is not described. While the measured data qualitatively correspond to the readouts associated with cardiac fibrosis, the quantitative correlations are lacking. The model of hypoxia/reoxygenation appears oversimplified, as this condition has an important timing component that is not captured. As this model is not critical to the main findings of the study, it should be either rationalized or eliminated.

2) The authors should discuss the selection of experimental parameters in more detail, in the context of cardiac physiology and pathology. For example, the reduced strain is a consequence of fibrosis, not its cause, and the explanation of how exactly the pathological changes were modeled would be of interest.

3) The model involves only cardiac fibroblasts, studied in isolation. It is unclear how much the measured responses of cardiac fibroblasts could be affected by the presence of other cells in the heart, both directly (e.g., contractility) and indirectly (paracrine signals).

4) Some of the figures and the table repeat the same data. Table 1 and Figure 4 are summaries. Figure 1, Figure 2 and Figure 3 contain heat maps that are summaries of the data in the bar charts. Figure 5 is technical description (would be better as supplementary). The data in Figure 1, Figure 2 and Figure 3 contain protein secretion data from a multiplex Luminex array – i.e. multiple readouts from a simple experiment but organised into separate figures. All of these data could be condensed and consolidated.

5) There is no information on myofibroblast differentiation (e.g. ICC of α-SMA) which would be interesting to know as myofibroblast phenotype is of key importance in regulating ECM turnover and cardiac remodeling.

6) The numbers of cell donors, independent experiments, and biological replicates should be specified for all experimental data.

7) The reference list lacks important information (e.g. volume and page numbers).

---

## [Author Response]

*Essential revisions:*

*The reviewers raise a number of concerns that must be adequately addressed before the paper can be further considered for publication in eLife. Some of the required revisions will likely require further experimentation within the framework of the presented studies and techniques.*

*1) The validation of the model is not described. While the measured data qualitatively correspond to the readouts associated with cardiac fibrosis, the quantitative correlations are lacking. The model of hypoxia/reoxygenation appears oversimplified, as this condition has an important timing component that is not captured. As this model is not critical to the main findings of the study, it should be either rationalized or eliminated.*

We agree with this comment and thank the reviewers for suggesting ways to improve the description of our approach in the context of current standard fibrotic models. Current literature shows how the most widely accepted in vitromodels for investigating cardiac fibroblasts responses to a fibrotic environment involve the stimulation of cultures with soluble pro-fibrotic or inflammatory cytokines (the most relevant being TGF-β, Angiotensin II and interleukins) (see Talman, et al., 2016; Fan, et al., 2012 referenced in the manuscript). We quantitatively correlated the outcomes of our model to others in the literature, selected by following these criteria: choosing only models involving early responses and showing data obtained from protein expression assays or immunofluorescence (except for data regarding cell proliferation). These comparisons are now listed in Table 2 supplementary new graph ([Supplementary-material SD4-data]) displays quantitative changes observed in standard and on-chip model according to the marker of interest. They allowed us to highlight pathological aspects that the model is efficiently capturing and behaviors that are not induced by tuning the microenvironment. This analysis revealed that CFs responses elicited in our model are in agreement with the ones observed in the existing models in both qualitative and quantitative terms. On the other hand, we also highlight that our model is the first to capture CF secretion of PDGF (a highly relevant factor in post-injury myocardial response), contrary to standard models. Also, we found that two inflammatory cytokines associated with cardiac injury and fibrosis (TNF-α and IL-6) are not upregulated by environmental stimuli, contrary to some reports that employ soluble pro-fibrotic stimulation. A fragment covering these points has been now added to the manuscript Discussion (tenth paragraph).

While we did not provide extensive quantitative correlations to in vivo data due to the dramatic differences in assay types (e.g. serum/plasma levels of markers) and experimental contexts, we highlighted interesting correlations with, for instance, early increased levels of MMPs or PDGF in patients (Discussion, tenth paragraph).

We also agree with the comment that our hypoxia-reperfusion model has limited impact on the main findings of our work. It was indeed previously included in the manuscript to serve as a demonstration of the capability of our novel microsystem. We have removed results of the hypoxia-reperfusion condition from the text and figures of the revised version of the manuscript.

*2) The authors should discuss the selection of experimental parameters in more detail, in the context of cardiac physiology and pathology. For example, the reduced strain is a consequence of fibrosis, not its cause, and the explanation of how exactly the pathological changes were modeled would be of interest.*

We agree with the reviewers about the lack of a proper context for the selection of the model parameters within the biological problem investigated. We added a paragraph to the Introduction covering in detail the choice of oxygen levels, mechanical strain levels and timings.

We agree with the reviewers that chronic cardiac fibrosis is the cause of myocardial scar formation and hence local reduced contractility. However, this event is the result of a slower tissue response, a culmination of the injury-related myocardial remodeling (inflammatory, proliferative and maturation phases) and occurring over a timeframe widely described as lasting weeks. In its current form and experimental exploitation, our model is not meant to account for and recapitulate such long-term phenomena in vitro. Rather, we focused on the fundamental cellular responses initiating the pathology. These responses have been shown (in our data and previous studies, see Turner, et al., 2007; Turner et al., 2009; Ugolini, et al., 2016, Van Nieuwenhoven, et al., 2013) to be evoked at early timeframes (in the acute phase of hours or days). in vivo human studies have also reported how, for instance, remodeling factors (e.g., MMPs) peak 24h post-injury (see Morishita, et al., 2015). We therefore revised the text in order to better describe our work as a model of early acute CF responses rather than a model of chronic cardiac fibrosis.

In this early and acute pathological context, it has been observed in patients that the myocardium undergoes abrupt mechanical changes immediately following injury (0-48h). Cardiac imaging studies agree in the interpretation that global strains are abruptly reduced shortly after myocardial insult (see Flachskampf et al., 2011; Hoit, 2011; Mollema et al., 2010). Quantitatively, strain values recorded depend on imaging algorithms and vary significantly throughout the heart. It is therefore difficult to directly translate them in vitro. The differences between healthy myocardial strains and ischemic/injured regions have been however reported to be 2-4 fold lower: Vartdal, et al., (2007) described an ischemic peak systolic strain of approximately 4% versus a physiological values of approximately 13%; Dandel, et al. (2009) report pathological values lower than 3% vs. physiological values of 10%. Our choice of mechanical strain values (2% strain, 8% strain and static control) was guided by such imaging studies and is meant to be in line with previous observations on CFs different responses to different strain regimes (Ugolini, et al., 2016) in order to provide comparable data.

In terms of oxygen levels, there is wide agreement on the fact that in vivohealthymyocardial tissue is characterized by an oxygen concentration of 5-6% (see Sen et al., 2006; Winegrad, et al., 1999; Gonschior, et al., 1992) and on the fact that upon ischemic cardiac injuries oxygen levels drop to 0-1% due to impaired blood flow (Roy, et al., 2003).

We hope to have better described our work in the context of cardiac physiology and pathology, particularly in the Introduction where a new paragraph addresses the above previous unclear points (Introduction, last paragraph).

*3) The model involves only cardiac fibroblasts, studied in isolation. It is unclear how much the measured responses of cardiac fibroblasts could be affected by the presence of other cells in the heart, both directly (e.g., contractility) and indirectly (paracrine signals).*

We thank the reviewers for suggesting a more detailed discussion of the missing components of the model compared to in vivoheart homeostasis. We added a more detailed paragraph in the Discussion (last paragraph; and reported below) that describes how the missing heart cellular actors could influence CF responses. Since our finely tuned mechanical stimulation system accounts for changes in contractility directly and uniformly delivered to cultured CFs, we believe indirect paracrine signals represent the key environmental factors that should be addressed by future studies (e.g., introducing a multi-culture in our model). Indeed, we made use of the comparison with other in vitromodels that employ soluble signals to discuss how additional pathological responses of CFs could be elicited by soluble factors provided by nearby apoptotic cardiac myocytes or infiltrated immune cells.

“This last missing readout suggests that indirect paracrine action provided by other cellular actors involved in myocardial homeostasis and typically modeled in standard biochemical stimulation approaches is likely to play a role in driving those CF pathological behaviors that are not captured in our model. […] Consequently, improving the current model with the addition of indirect paracrine action by other cellular actors is a potentially promising future direction for this study, expected to lead to a full on-chip recapitulation of early cardiac injury cellular events by building solely on a fine mimicry of the microenvironment rather than on current biochemical stimulation approaches.”

*4) Some of the figures and the table repeat the same data. Table 1 and Figure 4 are summaries. Figure 1, Figure 2 and Figure 3 contain heat maps that are summaries of the data in the bar charts. Figure 5 is technical description (would be better as supplementary). The data in Figure 1, Figure 2 and Figure 3 contain protein secretion data from a multiplex Luminex array – i.e. multiple readouts from a simple experiment but organised into separate figures. All of these data could be condensed and consolidated.*

We thank the reviewers for highlighting the excessive redundancy in the presentation of our data. We agree with this comment and performed substantial changes aimed at condensing the information presented in the manuscript. Original Figure 4 (radar graph) was removed from the manuscript and we kept only Table 1 as we found it a useful summary of results and experimental conditions. All heat maps were removed from the manuscript. We condensed Figure 3 and Figure 3—figure supplement 1 into one Figure 3. The technical inset in Figure 5 was moved in a supplemental figure, but we opted for maintaining the experimental design as a main Figure since we believe it could be helpful as further clarification of the experimental parameters and timeline. We also opted for keeping the Luminex data separated into multiple figures since the entire Results section is subdivided according to the main cellular responses rather than the assays performed. We are open to modifying the manuscript structure if the reviewers still feel that this represents a critical issue for the presentation of data.

*5) There is no information on myofibroblast differentiation (e.g. ICC of α-SMA) which would be interesting to know as myofibroblast phenotype is of key importance in regulating ECM turnover and cardiac remodeling.*

We agree with the reviewers on the importance of myofibroblast differentiation in the context of cardiac injury and remodeling. We therefore performed additional experiments aimed at determining expression and localization of αSMA in CFs cultured in all our combined conditions of oxygen levels and mechanical stimulation. These new results have been added to a new paragraph in Results (paragraph “D. Myofibroblast differentiation”; and new Figure 4 in the revised version) and reveal that a diffuse cytoplasmic αSMA staining is present in CFs cultured under all conditions. The expressed αSMA does not get incorporated into CFs stress fibers, as revealed by actin counterstaining, indicating that the early environmental conditions are not sufficient to induce myofibroblast differentiation per se. In addition, we included positive controls for our stainings (CFs supplemented with high levels of TGF-β, a known inducer of myofibroblast differentiation) that show how CFs exhibit αSMA co-localized with actin stress fibers.

*6) The numbers of cell donors, independent experiments, and biological replicates should be specified for all experimental data.*

We added this information to each figure caption for all experimental data collected.

*7) The reference list lacks important information (e.g. volume and page numbers).*

We have fixed the errors in the references.